# STREAK: A supervised cell surface receptor abundance estimation strategy for single cell RNA-sequencing data using feature selection and thresholded gene set scoring

**Azka Javaid**[ID]*, **Hildreth Robert Frost**

Department of Biomedical Data Science, Dartmouth College, Hanover, New Hampshire, United States of America

* azka.javaid.gr@dartmouth.edu

**Data Availability Statement:** All relevant data are within the manuscript and its Supporting information files. Please see our Github repository

## Abstract

The accurate estimation of cell surface receptor abundance for single cell transcriptomics data is important for the tasks of cell type and phenotype categorization and cell-cell interaction quantification. We previously developed an unsupervised receptor abundance estimation technique named SPECK (Surface Protein abundance Estimation using CKmeans-based clustered thresholding) to address the challenges associated with accurate abundance estimation. In that paper, we concluded that SPECK results in improved concordance with Cellular Indexing of Transcriptomes and Epitopes by Sequencing (CITE-seq) data relative to comparative unsupervised abundance estimation techniques using only single-cell RNA-sequencing (scRNA-seq) data. In this paper, we outline a new supervised receptor abundance estimation method called STREAK (gene Set Testing-based Receptor abundance Estimation using Adjusted distances and cKmeans thresholding) that leverages associations learned from joint scRNA-seq/CITE-seq training data and a thresholded gene set scoring mechanism to estimate receptor abundance for scRNA-seq target data. We evaluate STREAK relative to both unsupervised and supervised receptor abundance estimation techniques using two evaluation approaches on six joint scRNA-seq/CITE-seq datasets that represent four human and mouse tissue types. We conclude that STREAK outperforms other abundance estimation strategies and provides a more biologically interpretable and transparent statistical model.

## Author summary

Herein, we present an overview of our recently developed supervised receptor abundance estimation technique, STREAK (gene Set Testing-based Receptor abundance Estimation using Adjusted distances and cKmeans thresholding), which leverages co-expression associations learned from joint scRNA-seq/CITE-seq data to perform approximate abundance estimation. More specifically, STREAK functions by utilizing these expression associations to develop weighted membership gene sets, which are next thresholded following a

for the associated STREAK package at [https://github.com/azkajavaid/STREAK].

**Funding:** This work was funded by National Institutes of Health grants R35GM146586, R21CA253408, P20GM130454 and P30CA023108. The funders had no role in study design, data collection and analysis, decision to publish, or preparation of the manuscript. No authors received a salary from any of the funders.

**Competing interests:** The authors have declared that no competing interests exist.

gene set scoring procedure. These thresholded scores are set to the estimated abundance profiles.

We validate STREAK relative to both unsupervised and supervised estimation approaches using two different evaluation approaches, which include a cross-validation and a cross-training strategy, and approximately four different tissue types, which include the peripheral blood mononuclear cells, mesothelial cells, monocytes and lymphoid tissue. We conclude that STREAK outperforms comparative receptor abundance estimation approaches via a relatively more biologically interpretable and transparent statistical model, facilitated by VAM's (the Variance-adjusted Mahalanobis distance measure) customizable gene set scoring procedure.

## Introduction

Single cell RNA-sequencing (scRNA-seq) technologies, such as the 10X Chromium system [1], can now cost effectively profile gene expression in tens-of-thousands of cells dissociated from a single tissue sample [2, 3]. The transcriptomic data captured via scRNA-seq gives researchers unprecedented insight into the cell types and phenotypes that comprise complex tissues such as the tumor microenvironment or brain. Single cell transcriptomics can also facilitate the characterization of cell-cell signaling [4] and the identification of co-regulated genetic modules and gene-regulatory networks [5]. A critical element of these cell type/phenotype identification and cell-cell interaction analysis tasks is the accurate estimation of receptor protein abundance. Although direct receptor protein measurements are sometimes available, via either fluorescence-activated cell sorting (FACS)-based [6] enrichment prior to scRNA-seq analysis or through joint scRNA-seq/CITE-seq (Cellular Indexing of Transcriptomes and Epitopes by Sequencing) [7] profiling, most single cell data sets capture only gene expression values. For such data, receptor abundance estimates can be generated using either a supervised method (i.e., a method that leverages associations learned on joint transcriptomic/proteomic training data) or an unsupervised method (i.e., a method that generates estimates from the target scRNA-seq data without reference to a trained model).

A common unsupervised approach for estimating receptor abundance uses the expression of the associated mRNA transcript as a proxy for the receptor protein. While this approach is plausible, it often results in low quality estimates given the significant sparsity of scRNA-seq data [8]. This sparsity problem is well illustrated by scRNA-seq data for FAC sorted immune cells from Zheng et al. [1], which found that a large fraction of cells have no detectable expression of the transcript whose corresponding receptor is positively expressed in those cells, e.g., only 20% of CD19+ B cells expressed the CD19 transcript.

To overcome scRNA-seq sparsity, the standard bioinformatics workflow uses the average expression of the receptor transcript across large populations or cluster of cells to estimate receptor protein abundance. Although a cluster-based analysis mitigates sparsity, it has two major limitations. First, it assumes that receptor abundance is uniform across all cells in a given cluster, ignoring potentially significant within cluster heterogeneity. Second, it generates only a small number of independent receptor abundance estimates (one per cluster), which limits insight into the joint distribution of different receptors. A key benefit of single cell transcriptomics is the dramatic increase in sample size, with each cell providing a distinct expression profile. These large sample sizes can significantly improve estimates of the marginal and joint distribution of gene expression values. To support the cluster-free estimation of receptor abundance, we recently developed the Surface Protein abundance Estimation using CKmeans-

based clustered thresholding (SPECK) method. SPECK uses a low-rank reconstruction of scRNA-seq data followed by a clustering-based thresholding of the reconstructed gene expression values to generate non-sparse estimates of receptor transcripts. While the SPECK approach significantly outperforms both the naïve approach of directly using the receptor transcript and comparative reduced rank reconstruction (RRR)-based abundance estimation strategies, accuracy is poor for several biologically important receptors, e.g., receptors such as CD69 for which protein abundance is not closely correlated with transcript expression [9]. For such receptors, a supervised approach is needed that can leverage joint transcriptomic/proteomic data to identify the gene expression signature that is most closely associated with protein abundance.

Existing supervised methods for receptor abundance estimation include cTP-net (single cell Transcriptome to Protein prediction with deep neural network) [10] and PIKE-R2P (Protein–protein Interaction network-based Knowledge Embedding with graph neural network for single-cell RNA to Protein prediction) [11]. cTP-net uses a multiple branch deep neural network (MB-DNN) on joint scRNA-seq/CITE-seq training data to generate cell-level surface protein abundance estimates for target scRNA-seq data. While the abundance estimates generated using cTP-net are highly correlated with corresponding protein measurements, only select 24 immunophenotype markers/receptors are currently supported. Additionally, since cTP-net uses a deep learning model trained on the same immune cell populations (i.e., peripheral blood mononuclear cells (PBMC), cord blood mononuclear cells (CBMC) and bone marrow mononuclear cells (BMMC)) to perform estimation via transfer learning, it may not be able to capture gene expression patterns that are specific to an individual dataset or that more broadly generalize to non-immune-related surface markers. PIKE-R2P uses a protein-protein interactions (PPI)-based graph neural network (GNN) integrated with prior knowledge embeddings to estimate receptor abundance values. This method assumes that since gene expression regulation mechanisms are likely shared between proteins, such similarities can be used to generate protein-protein interactions that can then be leveraged in a GNN. Like cTP-net, PIKE-R2P is limited to an analysis and assessment of model prediction results on a small group of 10 receptors. A second important limitation of neural network-based models such as PIKE-R2P and cTP-net is that the weights retrieved via the training process are generally not sufficiently transparent or biologically interpretable enough to be manually fine-tuned by medical practitioners.

Given the performance issues with unsupervised methods like SPECK and the limited support, generalizability, and applicability of existing supervised estimation techniques such as cTP-net, alternative receptor abundance estimation methods are needed. Here, we propose a novel supervised approach for cell-level receptor abundance estimation for scRNA-seq data named STREAK (gene Set Testing-based Receptor abundance Estimation using Adjusted distances and cKmeans thresholding) that leverages co-expression associations learned from joint scRNA-seq/CITE-seq training data to perform thresholded gene set scoring on target scRNA-seq data. We validate STREAK on six datasets representing about five different tissue types and compare its performance against both unsupervised receptor abundance estimation techniques, such as SPECK and normalized RNA transcript counts, and supervised methods. In addition to supervised receptor abundance estimation techniques that leverage trained associations learned from separate training datasets like cTP-net, we also evaluate STREAK against the Random Forest (RF) and Support Vector Machines (SVM) algorithms that are trained on a small separate subset of cells from the same joint scRNA-seq/CITE-seq training data as the target data. This latter supervised comparison with the RF and the SVM model is motivated by a recent study that analyzed performance of tree-based ensemble methods and neural networks and found RF to have a superior performance over neural nets for the task of receptor

abundance estimation [12]. We do not perform comparison against PIKE-R2P since, while an initial implementation exists on Github, the upstream package dependencies are not adequately documented. We evaluate all comparisons between the estimated receptor abundance profiles and CITE-seq ADT data using the Spearman rank correlation coefficient. Overall, we observe that STREAK has superior performance relative to comparative methods on the six analyzed datasets, which highlights the accuracy and generalizability of our proposed estimation approach. Moreover, since it allows the specification of custom gene weights for subsequent gene set scoring, STREAK has tremendous clinical utility as an interpretable and adaptable receptor abundance estimation strategy.

# Materials and methods

## STREAK method overview

STREAK performs receptor abundance estimation for target scRNA-seq data using trained associations learned from joint gene expression and protein abundance data (see Fig 1). These associations are used to construct weighted receptor membership gene sets with the set for each receptor containing genes whose normalized and reconstructed scRNA-seq expression values are most strongly correlated with CITE-seq protein abundance (see Algorithm 1). The weighted gene sets are next leveraged in the gene set scoring step to produce cell-specific scores for each receptor. Lastly, a thresholding mechanism is applied to the resulting scores and the estimated abundance estimates are set to this thresholded output (see Algorithm 2).

**Algorithm 1** STREAK Algorithm (Receptor gene set construction)
**Require:** $X_{tR} \in \mathbb{R}^{m_1 \times n}$—scRNA-seq training counts

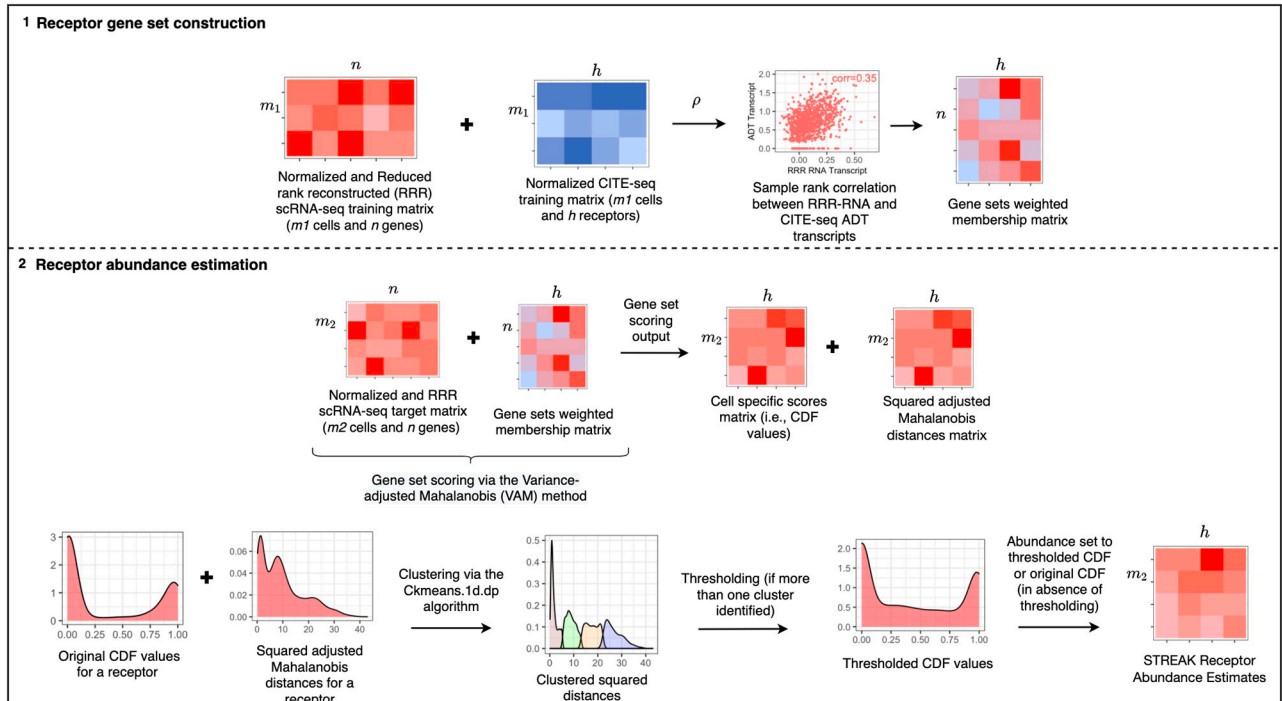

**Fig 1.** STREAK schematic with step (1) corresponding to the training co-expression analysis and step (2) corresponding to gene set scoring and subsequent clustering and thresholding to achieve cell-specific estimated receptor abundance profiles.

**Require:** $X_{tP} \in \mathbb{R}^{m_1 \times h}$—CITE-seq training counts
**Ensure:** $A \in \mathbb{R}^{n \times h}$—Gene sets weighted membership matrix
 *Normalization and RRR of training data*
1: $X_{tR} \xrightarrow{\text{Log-Normalization+RRR}} X_{tR}^*; X_{tP} \xrightarrow{\text{CLR-Normalization}} X_{tP}$
 *Co-expression analysis*
2: **for** $i \leftarrow 1$ to $h$ **do** ▷ Associations between scRNA-seq and CITE-seq
 training data
3: **for** $j \leftarrow 1$ to $n$ **do**
4: $r_x \leftarrow rank(X_{tR}^*[, j])$
5: $r_y \leftarrow rank(X_{tP}[, i])$
6: $A \leftarrow \frac{cov(r_x, r_y)}{\sigma_{r_x} \sigma_{r_y}}$ ▷ Pearson correlation on rank data (i.e., Spearman
 rank correlation)
7: **end for**
8: **end for**

 **Algorithm 2** STREAK Algorithm (Receptor abundance estimation)

**Require:** $X_R \in \mathbb{R}^{m_2 \times n}$—scRNA-seq target counts
**Require:** $A \in \mathbb{R}^{n \times h}$—Gene sets weighted membership matrix
**Ensure:** $S_R \in \mathbb{R}^{m_2 \times h}$—Estimated abundance profiles
 *Normalization and RRR of target data*
1: $X_R \xrightarrow{\text{Log-Normalization+RRR}} X_R^*$
 *Gene set scoring and thresholding*
2: **for** $k \leftarrow 1$ to $h$ **do**
3: topGenes $\leftarrow$ *names*($A[1:10, k]$)
4: topGenesWeights $\leftarrow A[1:10, k]$
5: testRRR $\leftarrow X_R^*[, \text{topGenes}]$ ▷ Top 10 most co-expressed genes
6: vamOut $\leftarrow$ *vam*(testRRR, *gene.weights* = topGenesWeights, *gamma* = T,
 *center* = F) ▷ Gene set scoring
7: vamCDF $\leftarrow$ vamOut["*cdf.value*"]
8: vamSqDist $\leftarrow$ vamOut["*distance.sq*"]
9: ckRes $\leftarrow$ *Ckmeans*.1*d.dp*(vamSqDist, $k$ = $c(1:4)$) ▷ Clustering
 of cell-specific squared distances
10: numCluster $\leftarrow$ ckRes["*cluster*"]
11: valCenters $\leftarrow$ ckRes["*centers*"]
12: **if** *length*(valCenters) $> 1$ **then**
13: minVal $\leftarrow$ *which*(valCenters = *min*(valCenters))
14: minNum $\leftarrow$ *which*(numCluster = minVal)
15: vamCDF[minNum] $\leftarrow 0$ ▷ Thresholding of cell-specific gene
 set scores
16: **end if**
17: $S_R \leftarrow$ vamCDF
18: **end for**

## Receptor gene set construction

In order to generate weighted gene sets for all supported receptors, we performed co-expression analysis on joint scRNA-seq/CITE-seq training data. It is important to note, however, that the weighted gene sets used by STREAK can be generated using alternative approaches, e.g., manually specified based on prior biological knowledge. The first step in the training process used for this paper generated $\mathbf{X}_{tR}^*$, a rank-$k$ reconstruction of the $m_1 \times n$ training scRNA-seq matrix $\mathbf{X}_{tR}$ holding the log-normalized gene expression counts for $n$ genes in $m_1$ cells. The RRR of the scRNA-seq data was performed using the randomized SVD algorithm from the rsvd R package [13] based on the rank selection procedure proposed in the SPECK paper [9], which utilizes the rate of change in the standard deviation of the non-centered sample principal components to compute the estimated rank-$k$. The analogous $m_1 \times h$ CITE-seq counts

matrix $\mathbf{X}_{tP}$ from the joint scRNA-seq/CITE-seq training data was normalized using the centered log-ratio (CLR) transformation. Both the log-normalization and CLR transformation were performed using Seurat [14–17]. Following normalization and RRR, the $\mathbf{X}_{tR}^*$ and $\mathbf{X}_{tP}$ matrices were used to define gene sets for each of the $h$ supported receptors, with the set for each receptor containing the genes whose reconstructed expression values from $\mathbf{X}_{tR}^*$ have the largest positive Spearman rank correlation with the corresponding receptor's normalized CITE-seq values from $\mathbf{X}_{tP}$. The rank correlation values were also used to define positive weights for the genes in each set. For the results reported in this paper, a gene set size of 10 was used.

## Receptor abundance estimation

**Single cell gene set scoring.**  Given weighted gene sets for $h$ target receptors, we used a thresholded single cell gene set scoring mechanism to generate receptor abundance estimates. For a $m_2 \times n$ matrix $X_R$ that holds scRNA-seq data for $m_2$ cells and $n$ genes, we first performed normalization and RRR to output the $m_2 \times n$ matrix $\mathbf{X}_R^*$. We next computed cell-level scores for each of the $h$ target receptors using $\mathbf{X}_R^*$ with the Variance-adjusted Mahalanobis (VAM) gene set scoring [18] method.

Execution of the VAM method requires two input matrices:

1. $\mathbf{X}_R^*$: a $m_2 \times n$ scRNA-seq target matrix containing the positive normalized and RRR counts for $n$ genes in $m_2$ cells.

2. $\mathbf{A}$: a $n \times h$ matrix that captures the weighted annotation of $n$ genes to $h$ gene sets, i.e., gene sets for each of the $h$ receptor proteins. If gene $i$ is included in gene set $j$, then element $a_{i,j}$ holds the gene weight, otherwise, $a_{i,j}$ is not defined.

VAM outputs a $m_2 \times h$ matrix, $\mathbf{M}$, that holds the cell-specific squared modified Mahalanobis distances for $m_2$ cells and $h$ gene sets, and a $m_2 \times h$ matrix, $\mathbf{S}$, that holds the cell-level scores for $m_2$ cells and $h$ gene sets. The computation for both matrices $\mathbf{M}$ and $\mathbf{S}$ is detailed below (see the VAM paper [18] for additional details).

1. **Technical variances estimation**: The length $n$ vector $\sigma_{tech}^2$ holding the technical variance of each gene in $\mathbf{X}_R^*$ is first computed using the Seurat variance decomposition approach for either the log-normalized or SCTransform-normalized data [19]. Alternatively, the elements of $\sigma_{tech}^2$ can be set to the sample variance of each gene in $\mathbf{X}_R^*$ under the assumption that the observed marginal variance of each gene is entirely technical.

2. **Modified Mahalanobis distances computation**: Given $\mathbf{M}$, a $m_2 \times h$ matrix of squared values of a modified Mahalanobis distance, each column $k$ of $\mathbf{M}$, which holds the cell-level squared distances for gene set $k$, is calculated as $M[,k] = diag(X_k(I_g \sigma_{g,tech}^2)^{-1} X_k^T)$ where $g$ corresponds to the size of the gene set $k$, $X_k$ is a $m_2 \times g$ matrix containing the $g$ columns of $\mathbf{X}_R^*$ relating to the members of set $k$ (i.e., the $g$ elements of column $k$ of $\mathbf{A}$ with non-zero values), $I_g$ is a $g \times g$ identity matrix, and $\sigma_{g,tech}^2$ contains the elements of $\sigma_{tech}^2$ associated with the $g$ genes in set $k$. To prioritize genes with large weights, the elements of the $\sigma_{g,tech}^2$ vector are divided by the corresponding elements of column $k$ of $\mathbf{A}$. This modification will shrink the effective variance for genes with large weights resulting in a larger Mahalanobis distance. Modified Mahalanobis distances are additionally recomputed on a version of $\mathbf{X}_R^*$ where the row labels of each column are randomly permuted, which captures the distribution of the squared modified Mahalanobis distances under the $H_0$ that the normalized and RRR expression values in $\mathbf{X}_R^*$ are uncorrelated with only the technical variance. $\mathbf{X_P}$ thereby

represents the row-permuted version $\mathbf{X}_R^*$ and $\mathbf{M_P}$ represents the $m_2 \times h$ matrix that contains the squared modified Mahalanobis distances computed on $\mathbf{X_P}$.

3. **Gamma distribution fit**: A gamma distribution is individually fit to the non-zero elements in each column of $\mathbf{M_P}$ using the method of maximum likelihood. Alternatively, gamma distributions can be fit directly on $\mathbf{M}$ to alleviate the computational costs of generating $\mathbf{X_P}$ and $\mathbf{M_P}$.

4. **Cell-specific scores computation**: Cell-level gene set scores, matrix $\mathbf{S}$, are defined to be the gamma cumulative distribution function (CDF) value for each element of $\mathbf{M}$.

**One-dimensional clustering and thresholding.**   Following the generation of cell-specific squared distances, matrix $\mathbf{M}$, and cell-specific gene set scores, matrix $\mathbf{S}$, we used the *Ckmeans.1d.dp* algorithm from Ckmeans.1d.dp v3.3.3 [20, 21] to perform one-dimensional clustering on $\mathbf{M}$. Each column $k$ of $\mathbf{M}$, which holds the cell-specific squared distances for gene set $k$, was clustered with the number of computed clusters bound between one and four. If more than one cluster was identified, then all the non-zero cell-specific gene set scores corresponding to the indices of the least-valued cluster for $k$ from the analogous matrix $\mathbf{S}$ were set to zero. All zero values corresponding to the indices of the least-valued cluster and the remaining non-zero and zero values corresponding to the indices of the higher-valued clusters for $k$ were retained. If only one cluster was identified for gene set $k$ of $\mathbf{M}$, then thresholding was not performed and the analogous cell-specific gene set scores for gene set $k$ of $\mathbf{S}$ were preserved as estimated abundance profiles.

## Evaluation

**Datasets.**   Comparative evaluation of STREAK was performed on six publicly accessible joint scRNA-seq/CITE-seq datasets generated on approximately four human and one mouse tissue types: 1) the Hao et al. [14] human PBMC dataset (GEO [22] series GSE164378) contains 161,764 cells profiled using 10X Chromium 3' with 228 TotalSeq A antibodies, 2) the Unterman et al. [23] human PBMC dataset (GEO series GSE155224) contains 163,452 cells profiled using 10X Chromium 5' with 189 TotalSeq C antibodies, 3) the 10X Genomics [24] human extranodal marginal zone B-cell tumor/mucosa-associated lymphoid tissue (MALT) dataset contains 8,412 cells profiled using 10X Chromium 3' with 17 TotalSeq B antibodies, 4) the Lakkis et al. [25] human blood monocyte and dendritic cell dataset profiled with 238 antibodies, 5) the Ma et al. [26] malignant peritoneal mesothelioma (MPEM) dataset profiled with 46 antibodies and 6) the Gayoso et al. [27] mus musculus dataset (GSE150599) profiled with 102 mouse antibodies. Hao data was generated on PBMC samples obtained from eight volunteers enrolled in a HIV vaccine trial [28, 29], with age ranging from 20 to 49 years. In comparison, the PBMC samples for the Unterman data were obtained from 10 COVID-19 patients and 13 matched controls with a mean age of 71 years. The two PBMC datasets thus consisted of different underlying patient populations with varying age ranges. All six scRNA-seq/CITE-seq datasets were processed using Seurat v.4.1.0 [14–17] in R v.4.1.2 [30].

We performed method evaluation using two approaches. First, we used a 5-fold cross-validation strategy where we utilized a subset of cells from each of the six joint scRNA-seq/CITE-seq datasets as training data and a second distinct subset of the same dataset as target data. This approach is suitable for analytical scenarios where joint scRNA-seq/CITE-seq data is only captured for a subset of cells from the original, larger population of cells for which only scRNA-seq data is available. Given this first scenario, receptor abundance levels can be estimated by mapping associations learned on the smaller subset of cells with joint scRNA-seq/

CITE-seq training data to the larger population with only scRNA-seq data. Our second evaluation strategy used a cross-training approach to learn and evaluate associations on separate datasets. This latter approach, which was used to learn associations on the Hao data and evaluate on the Unterman data, is suitable for situations where co-expression patterns learned from one joint scRNA-seq/CITE-seq data can be leveraged to perform receptor abundance estimation for another disjoint dataset that is of the same tissue type as the first data but only contains quantified scRNA-seq expression profiles. For both the cross-validation and the cross-training approaches, concordance between the estimated receptor abundance values for the target scRNA-seq data and the analogous ADT transcripts from the same joint, target scRNA-seq/CITE-seq data was quantified using the Spearman rank correlation, which was used to measure relationships between the relative ranks of the estimated abundance values [31].

**Comparison methods.** STREAK was evaluated against existing unsupervised and supervised-learning-based receptor abundance estimation techniques. Aside from the normalized RNA transcript approach, SPECK was the only unsupervised receptor abundance estimation method evaluated since it previously performed better than the comparative unsupervised techniques MAGIC (Markov Affinity-based Graph Imputation of Cells) [32] and ALRA (Adaptively thresholded Low-Rank Approximation) [33]. All comparative approaches are detailed below.

- Normalized RNA transcript: RNA transcript associated with the scRNA-seq count matrix was normalized using Seurat's log-normalization procedure [14–17] and set as the estimated receptor abundance.

- SPECK: scRNA-seq count matrix was normalized, RRR and thresholded with the *speck* function from the SPECK v0.1.1 R package [34]. Receptor abundance was set to this estimated output.

- cTP-net: Estimation was performed on the scRNA-seq count matrix with the *cTPnet* function and default parameters using cTP-net v1.0.3 R package [10]. Receptor abundance was set to this estimated output. The scRNA-seq matrix was not denoised with the SAVER-X package since cTP-net maintainers note that cTP-net can predict protein abundance relatively accurately without denoising. Denoising using SAVER-X was additionally not performed due to the associated high time complexity for large datasets and pending Python package dependency updates required by the SAVER-X package maintainers.

- Random Forest (RF): A random forest model was trained using the *randomForest* function with default parameters from the randomForest v4.7-1.1 R package [35]. Model training leveraged the top 10 genes whose normalized expression values from the scRNA-seq component of the scRNA-seq/CITE-seq joint training data had the largest positive Spearman rank correlation with the receptor's normalized ADT values from the corresponding CITE-seq component of the same joint training data. This trained RF model was applied to the target scRNA-seq data to generate estimated receptor abundance values.

- Support Vector Machines (SVM): A SVM model was trained using the *svm* function with default parameters from the e1071 v1.7.13 R package [36]. Similar to the random forest approach detailed above, scRNA-seq expression from top 10 genes with normalized scRNA-seq expression values most correlated with corresponding ADT values from a joint scRNA-seq/CITE-seq dataset was applied to the target scRNA-seq data to generate measures of estimated receptor abundance.

**Benchmark setup.** Each of the cross-validation and cross-training evaluation approaches had a distinct benchmark setup. For the cross-validation approach, five individual subsets of

**Table 1. Joint scRNA-seq/CITE-seq datasets used for method evaluation.**

| Species | Source Tissue | Final Subset Size | Number of Antibodies |
|---------|---------------|-------------------|----------------------|
| Human | PBMC Hao [14] | 60,000 | 211 |
| Human | PBMC Unterman [23] | 60,000 | 167 |
| Human | MALT [Mucosa-Associated Lymphoid Tissue] [24] | 8,412 | 17 |
| Human | Monocytes [25] | 37,000 | 238 |
| Human | MPEM [Malignant Peritoneal Mesothelioma] [26] | 4,969 | 46 |
| Mouse | Spleen and Lymph Nodes [37] | 20,000 | 102 |

different sizes were selected from each dataset to get a 20–80 train-test split where 20% of total cells were used as training data and 80% of cells were used as test data. The number of cells used in the training and test subsets and the number of total cells (i.e., sum of cells from the training and target data subsets) for each dataset are indicated in Tables 1 and 2 for the cross-validation approach. Evaluation using the PBMC cross-training strategy was performed on gene sets trained on a subset of 5,000 cells from the Hao data. Each trained subset was evaluated on five subsets of 5,000, 7,000 and 10,000 cells and one subset of 50,000 cells from the Unterman data.

Lower and upper cell count limits for the cross-validation approach were individually determined for each dataset. For the Hao data, the upper limit of 60,000 total cells (i.e., 12,000 training cells and 48,000 target cells) was determined by the capacity to perform RRR on 16 CPU cores without any virtual memory allocation errors. For the Unterman data, the upper limit of 50,000 total cells (i.e., 10,000 training cells and 40,000 target cells) was determined by the total number of cells in the CITE-seq assay (50,438 cells). For the MALT data, the upper limit of 8,410 total cells (i.e., 1,682 training cells and 6,728 target cells) was similarly determined by the total number of cells in the analogous CITE-seq assay (8,412 cells). The upper limit of 50,000

**Table 2. Number of cells in individual training and target data subsets and total number of cells for each of the six datasets for the 5-fold cross-validation evaluation approach.**

| Dataset | Number of Training Cells | Number of Target Cells | Total Number of Cells |
|---------|--------------------------|------------------------|-----------------------|
| PBMC Hao | 1,000 | 4,000 | 5,000 |
| | 2,500 | 10,000 | 12,500 |
| | 5,000 | 20,000 | 25,000 |
| | 10,000 | 40,000 | 50,000 |
| | 12,000 | 48,000 | 60,000 |
| PBMC Unterman | 1,000 | 4,000 | 5,000 |
| | 2,500 | 10,000 | 12,500 |
| | 5,000 | 20,000 | 25,000 |
| | 10,000 | 40,000 | 50,000 |
| MALT | 1,000 | 4,000 | 5,000 |
| | 1,250 | 5,000 | 6,250 |
| | 1,500 | 6,000 | 7,500 |
| | 1,682 | 6,728 | 8,410 |
| Monocytes | 7,422 | 29,690 | 37,112 |
| MPEM | 994 | 3,975 | 4,969 |
| Spleen and Lymph Nodes | 3,940 | 15,758 | 19,698 |

target cells for the cross-training approach was determined by the total number of cells in the CITE-seq assay corresponding to the Unterman data (50,438 cells).

From the initial 228 antibodies included in the Hao data, antibodies mapping to multiple HGNC (HUGO Gene Nomenclature Committee) [38] symbols or antibodies with their HGNC symbols not present in the feature/gene names corresponding to the scRNA-seq matrix were removed. Final assessment was performed for 217 antibodies for each subset of the Hao data as evaluated with the 5-fold cross-validation strategy. From the initial 200 antibodies included in the Unterman data, antibodies mapping to multiple HGNC symbols and mouse/rat specific antibodies were removed, resulting in 168 antibodies. Since assessment was performed for a varying number of cells, antibodies not expressed in smaller cell groups were dropped, resulting in assessment of either 167 or 168 antibodies for the Unterman data as evaluated with the cross-validation strategy. From the initial 17 antibodies included in the MALT data, three mouse/rat specific antibodies (IgG2a, IgG1 and IgG2b control) were removed. Final assessment was performed on 14 antibodies for the MALT data. For the cross-training approach, 124 antibodies, overlapping between the 217 antibodies from the Hao data and the 168 antibodies from the Unterman data, were assessed.

## Results

### STREAK generates receptor abundance estimates that are highly correlated with CITE-seq data

We first quantified the proportion of abundance profiles that have the highest Spearman rank correlation with the corresponding CITE-seq measurements when estimated using STREAK, SPECK or the normalized RNA method for the 5-fold cross-validation approach. We performed this computation for five different cell subsets of the Hao training and target data ranging from 1,000 training and 4,000 target cells to 12,000 training and 48,000 target cells and for four subsets of training and target cells ranging from 1,000 training and 4,000 target cells to 10,000 training and 40,000 target cells for the Unterman data and from 1,000 training and 4,000 target cells to 1,682 training and 6,728 target cells for the MALT data. We visualized each set of proportion estimates using a collective figure for all datasets, including Hao, Unterman, MALT, Monocytes, MPEM and Mouse Spleen and Lymph Nodes datasets, combined as indicated by Figs 2 and 3 as well as via separate graphical representations for each dataset as displayed by S5A, S5B and S5C Fig for the Hao, Unterman and the MALT datasets, respectively. Overall, the results indicate that the percentage of receptors for which the STREAK estimates have high rank correlations with analogous CITE-seq data are considerably larger than the corresponding percentage for SPECK or the normalized RNA approach for each of the Hao, Unterman and MALT datasets.

In addition to comparisons against existing unsupervised receptor abundance estimation strategies as displayed by Fig 4, we compared STREAK against supervised abundance estimation methods such as cTP-net and the RF model and similarly quantified the proportion of receptors for which the estimated abundance generated by each technique was most highly correlated with CITE-seq data. We similarly quantify comparisons for all datasets combined, as shown by Figs 5 and 6, as well as for individual data. S5D–S5F Fig present comparisons between STREAK and cTP-net while S5G–S5I Fig display comparisons between STREAK and the RF model for the Hao, Unterman and MALT datasets, respectively. Both sets of figures evidently show that a large percentage of abundance profiles estimated using STREAK are most highly correlated with corresponding CITE-seq data, which substantiates the claim that STREAK overall performs better than cTP-net and the RF model across all datasets and training subset sizes.

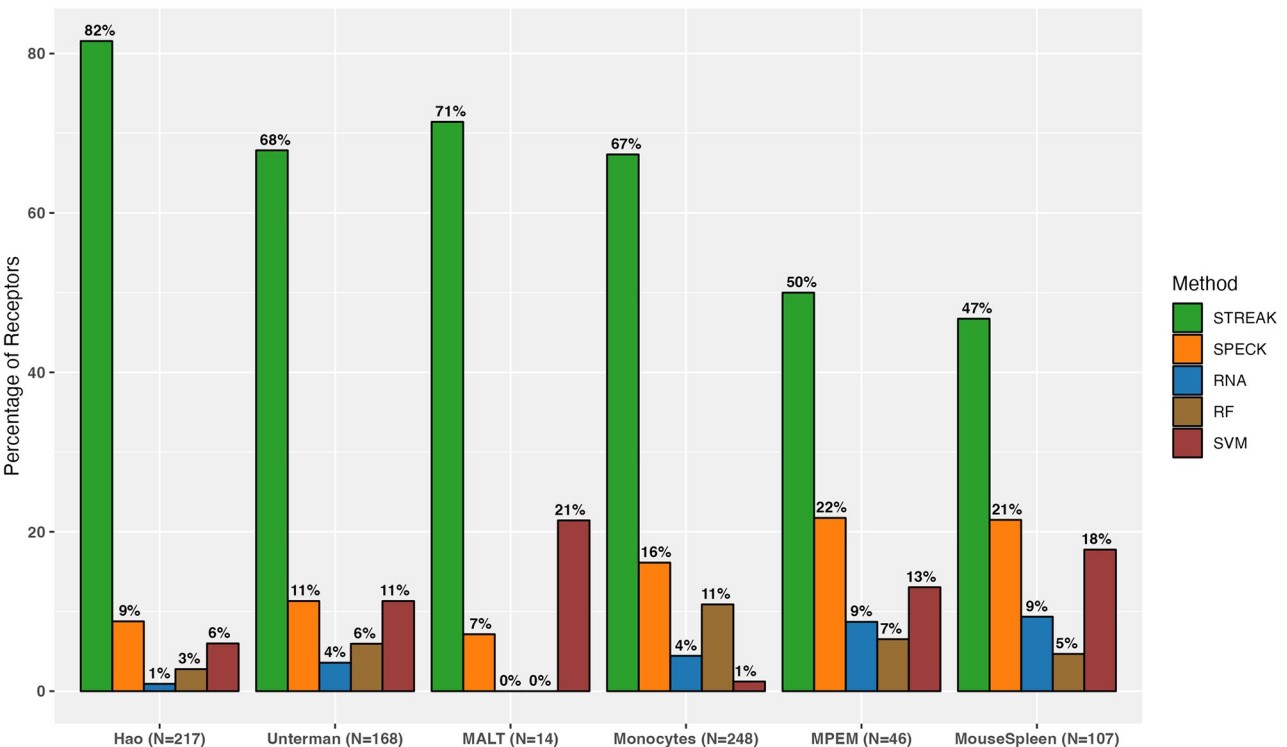

**Fig 2.** Percentage of receptors for which a given technique (STREAK, SPECK, normalized RNA transcript, Random Forest or Support Vector Machines) generates estimates with the highest average rank correlation with the associated CITE-seq data over five subsets of training data consisting of 12,000 cells from the Hao dataset, 10,000 cells from the Unterman data, 1,682 cells from the MALT data, 3,940 cells from the Mouse Spleen data, 7,422 cells from the Monocytes data and 994 cells from the MPEM dataset.

We next compared STREAK against unsupervised and supervised receptor abundance estimation approaches for the cross-training evaluation strategy, as indicated by Fig 7. We observed an equivalent pattern of improved concordance of the STREAK-based receptor abundance estimates with CITE-seq data relative to the estimates produced using SPECK, the normalized RNA approach, cTP-net, and the RF model. While the relative performance benefit of STREAK for the cross-training evaluation approach is more incremental than for the cross-validation strategy, the results nevertheless emphasize STREAK's ability to consistently generate more accurate abundance estimates than comparative methods. The fact that STREAK's relative performance is relatively insensitive to training and test dataset size across three main assessed datasets (PBMC Hao, PBMC Unterman and the 10X Genomics MALT) and two evaluation strategies further stresses the robustness of this approach for receptor abundance estimation.

In addition to evaluating STREAK against comparative methods using the evaluation strategy that assesses the proportion of receptors that are most correlated with CITE-seq data when estimated using STREAK, SPECK, normalized RNA transcript, the cTP-net approach and the RF and SVM algorithms (as presented using bar charts), we evaluated STREAK using correlation-correlation scatterplots. Figs 8 and 9 display the indicated percentage of receptors that are above the diagonal ($y = x$) line for STREAK versus the remaining comparative approaches for the PBMC Hao and the Mouse Spleen datasets. Overall, we observed that a large percentage of receptors are above the $y = x$ line in all of the subplots indicating that STREAK provides more accurate receptor abundance estimates than the comparison methods. A similar pattern is

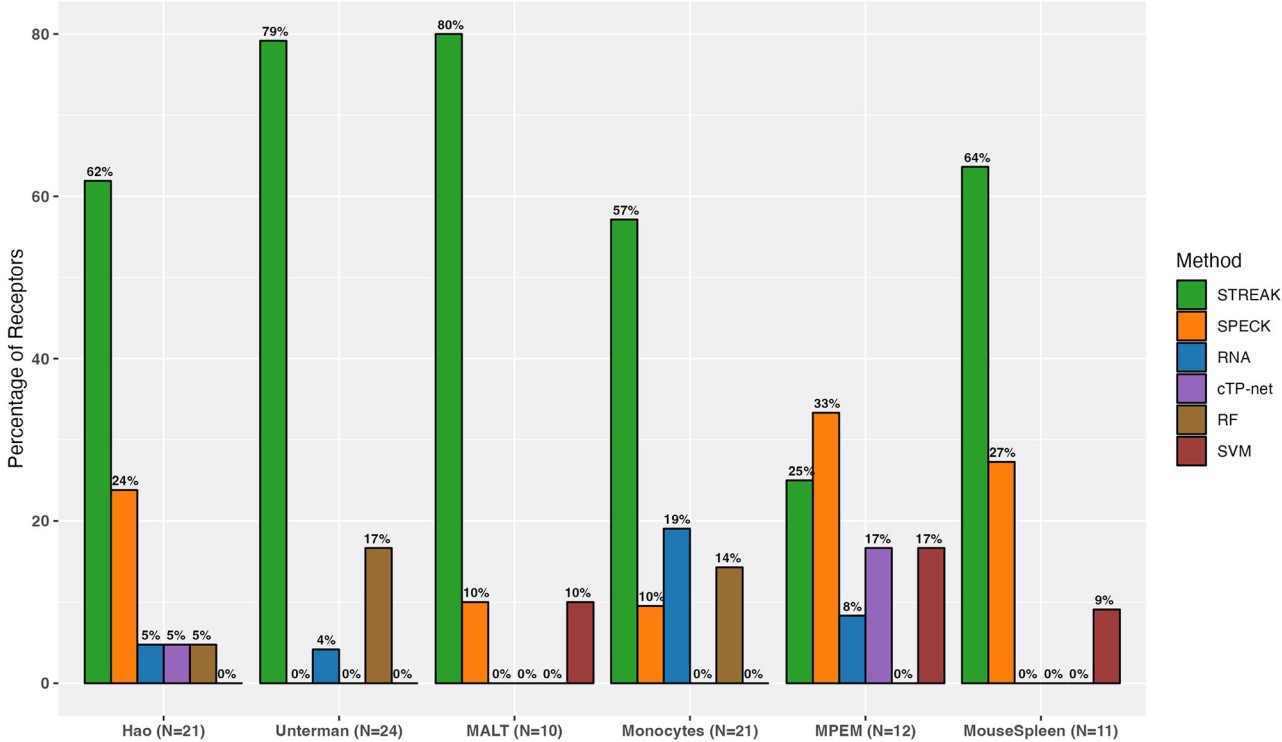

**Fig 3.** Percentage of receptors for which a given technique (STREAK, SPECK, normalized RNA transcript, Random Forest or Support Vector Machines) generates estimates with the highest average rank correlation with the associated CITE-seq data. Results are computed on the subset of receptors supported by the cTP-net methods.

found for the PBMC Unterman, MALT, Monocytes and MPEM datasets, as shown in S1–S4 Figs.

## STREAK as a tool for selecting the optimal abundance estimation strategy for specific receptors

Following an examination of overall rank correlation trends between CITE-seq data and corresponding abundance profiles estimated using STREAK and comparative methods across different training and test dataset sizes, we inspected individual trends in rank correlations over a select train/test split group for all six datasets, including the Hao, Unterman, MALT, Monocytes, MPEM and Mouse Spleen and Lymph Nodes. For the 5-fold cross-validation evaluation approach, we trained and evaluated abundance values estimated using STREAK, SPECK, the normalized RNA approach, cTP-net and the RF model using train/test splits of 12,000/48,000, 10,000/40,000, 1,682/6,728, 7,422/29,690, 994/3,975 and 3,940/15,758 cells for the Hao, Unterman, MALT, Monocytes, MPEM and Mouse Spleen and Lymph Nodes datasets, respectively. We visualized rank correlations between estimated abundance values and CITE-seq data using heatmaps. Asterisk text format was used to indicate receptors whose STREAK-based estimates have the highest rank correlation with analogous CITE-seq data in each plot. Figs 10 and 11 display these correlations between CITE-seq data and abundance profiles generated using all unsupervised and supervised estimation approaches (STREAK, SPECK, RNA, cTP-net, RF and SVM) for the Hao and the Mouse Spleen and Lymph Nodes datasets, respectively. These figures, along with the corresponding S13–S16 Figs, show that the abundance values estimated

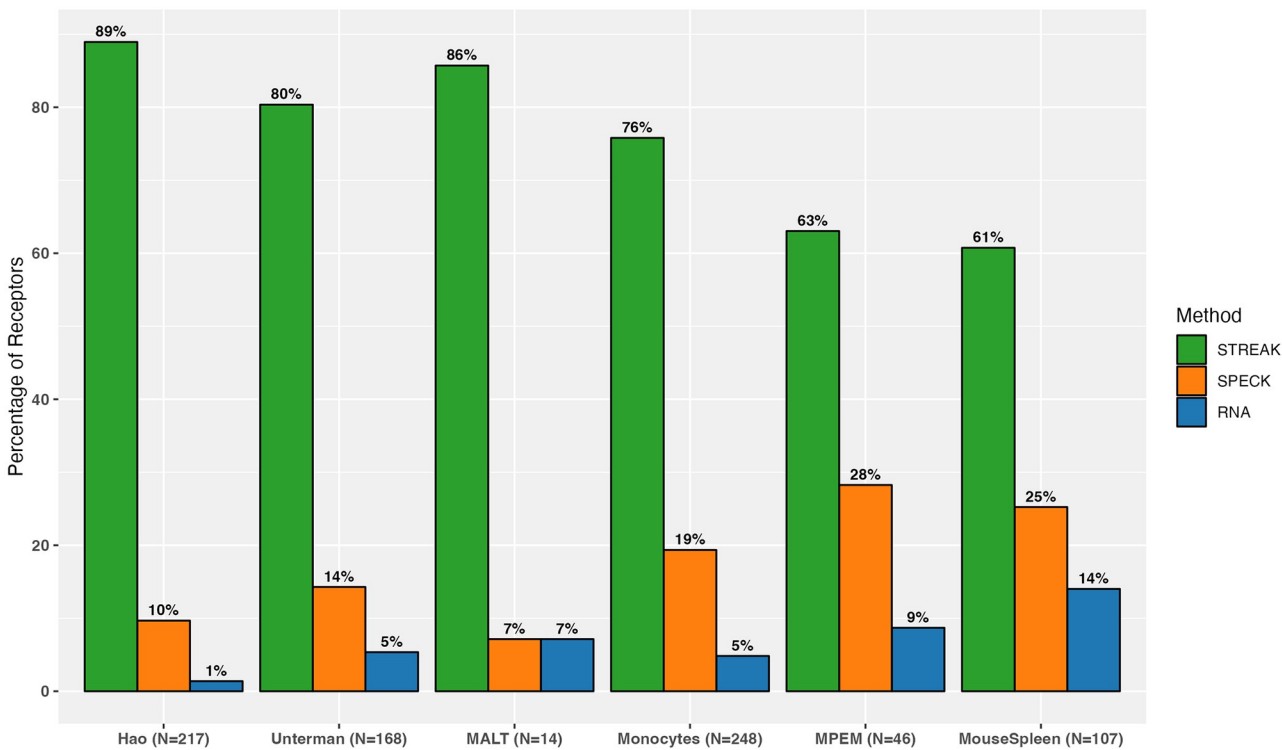

**Fig 4.** Percentage of receptors for which a given unsupervised abundance estimation technique (STREAK, SPECK and the normalized RNA transcript) generates estimates with the highest average rank correlation with the associated CITE-seq data.

with STREAK have relatively higher individual rank correlations with analogous CITE-seq data compared to alternative estimation strategies.

We similarly visualized rank correlations between CITE-seq data and receptor abundance profiles estimated using co-expression associations learned from a subset of 5,000 cells from the Hao data and comparative unsupervised and supervised abundance estimation methods as evaluated on a subset of 50,000 cells from the Unterman data for the cross-training approach. These results, which are displayed in Fig 12, show that while STREAK's performance as evaluated using the 5-fold cross-validation approach is superior to its performance when using the cross-training approach, STREAK has significantly better performance than the unsupervised techniques for the cross-training approach. Thus, while it is optimal to use trained associations learned from a subset of the same joint scRNA-seq/CITE-seq dataset as used for target estimation, there is still utility to leveraging associations from an independent scRNA-seq/CITE-seq dataset measured on the same tissue type as the target data.

Along with the practical advantage of these correlation heatmaps as strategies for evaluating estimated receptor abundance values, they can be leveraged to help identify an optimal abundance estimation approach for specific receptors. As shown in the heatmaps, while STREAK provides the best estimate for a majority of receptors, it is not optimal in all cases. For example, the rank correlation computed between the estimated CD14 abundance profiles and the corresponding CITE-seq values for the Hao data is higher for SPECK ($\rho = 0.754$) than STREAK ($\rho = 0.744$) (see Fig 10). Similarly, the rank correlation between the CD301 abundance profiles and the corresponding CITE-seq values for the Hao data is higher than the normalized RNA transcript ($\rho = 0.105$) as compared to STREAK ($\rho = 0.078$). Similarly, when assessing STREAK

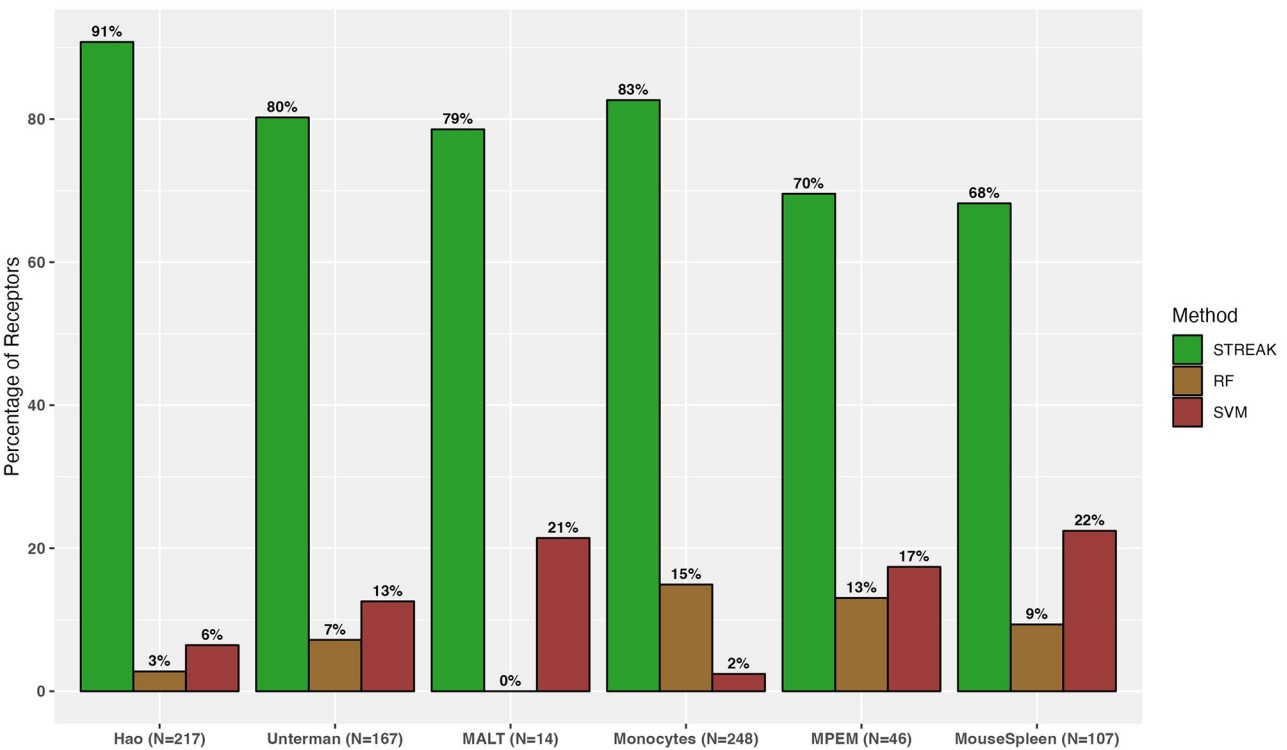

**Fig 5.** Percentage of receptors for which a given supervised abundance estimation technique (STREAK, the Random Forest and Support Vector Machines algorithms) generates estimates with the highest average rank correlation with the associated CITE-seq data.

against supervised abundance estimation techniques, the cTP-net-based profiles for the CD45RA receptor from the Hao data are more correlated with corresponding CITE-seq values ($\rho = 0.720$) than the STREAK-based abundance profiles ($\rho = 0.708$) (see Fig 10). These plots thus provide users with practical guidance when selecting the optimal abundance estimation approach for specific receptors.

## STREAK thresholding mechanism provides a considerable advantage over simple gene set scoring

Our next analysis evaluated the distinctive contributions of the gene set scoring and thresholding steps on STREAK's performance. For this purpose, we compared the gene set scores (i.e., the CDF values) generated directly by VAM with the thresholded VAM scores used by STREAK. Fig 13A, 13B and 13C indicate the proportion of receptors that have high rank correlations with analogous CITE-seq data when estimated using STREAK and VAM for the Hao, Unterman and MALT datasets, respectively, as evaluated using the 5-fold cross-validation approach. Overall, a greater percentage of receptors estimated with STREAK have higher rank correlations with CITE-seq values for all datasets and every training subset size, except the 1,000 cells training subset of the Hao data, versus direct use of the VAM scores. This result emphasizes the important contribution of thresholding in further improving the concordance of the estimated receptor abundance profiles with analogous CITE-seq data over and above VAM-based set scoring.

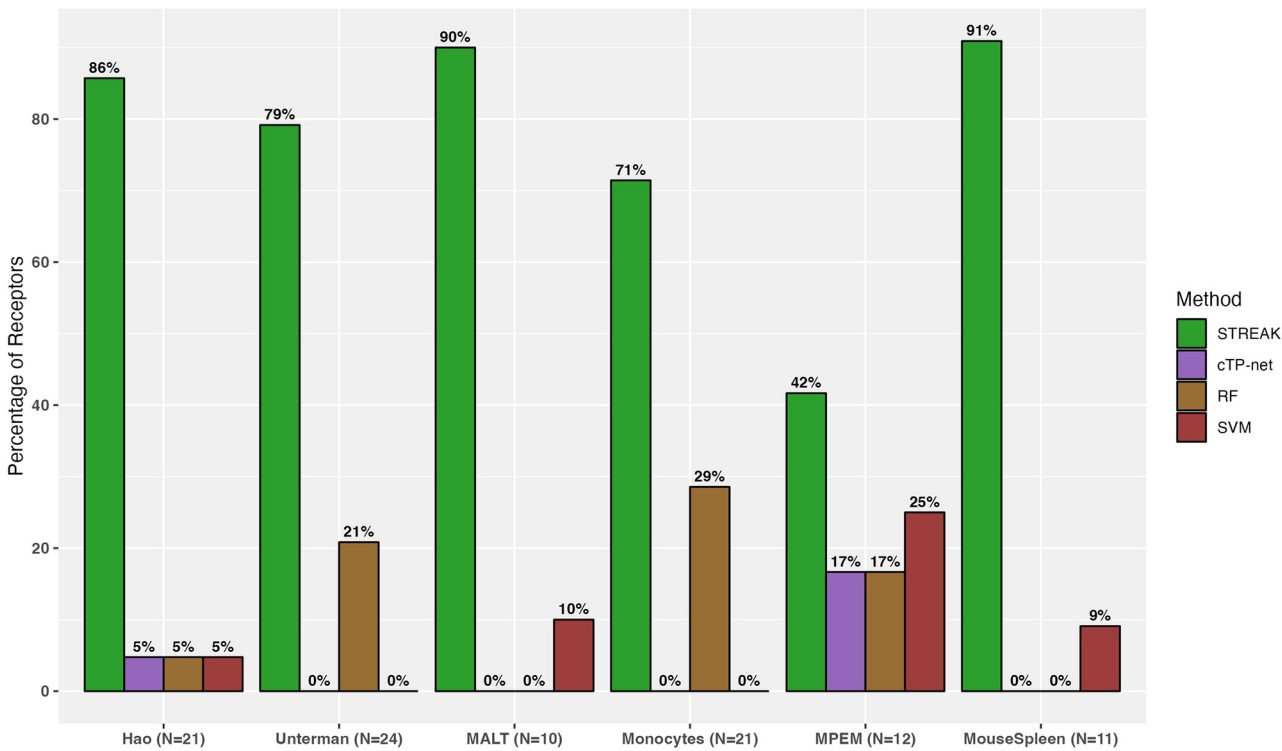

**Fig 6.** Percentage of receptors for which a given supervised abundance estimation technique (STREAK, cTP-net, and the Random Forest and Support Vector Machines algorithms) generates estimates with the highest average rank correlation with the associated CITE-seq data for a subset of receptors with defined cTP-net expression.

## STREAK is relatively insensitive to both training data subset size and gene set size

Following an assessment of the individual contributions of the gene set scoring and thresholding steps to STREAK performance, we implemented sensitivity analyses to examine the influence of training data size on STREAK's performance for the cross-training evaluation

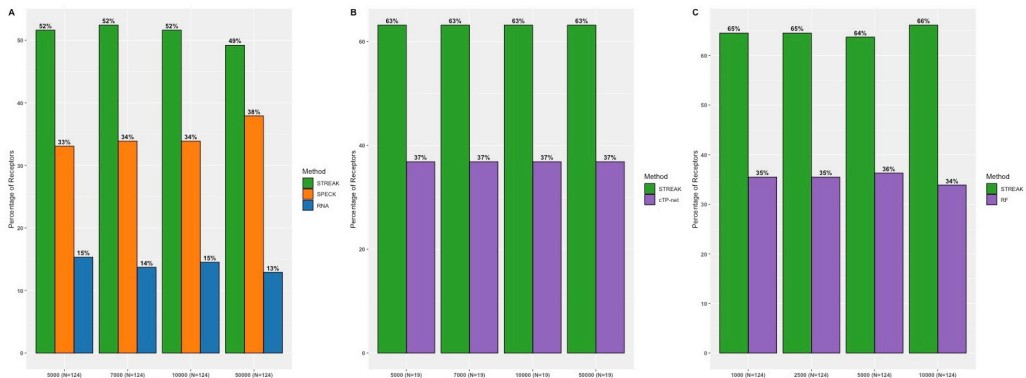

**Fig 7.** Percentage of receptors with highest average Spearman rank correlations between CITE-seq data and abundance profiles estimated using STREAK, SPECK, the normalized RNA approach, cTP-net or RF using the cross-training evaluation approach (Fig 1A–1C). The horizontal axis for these plots indicates the number of target cells evaluated from the Unterman data.

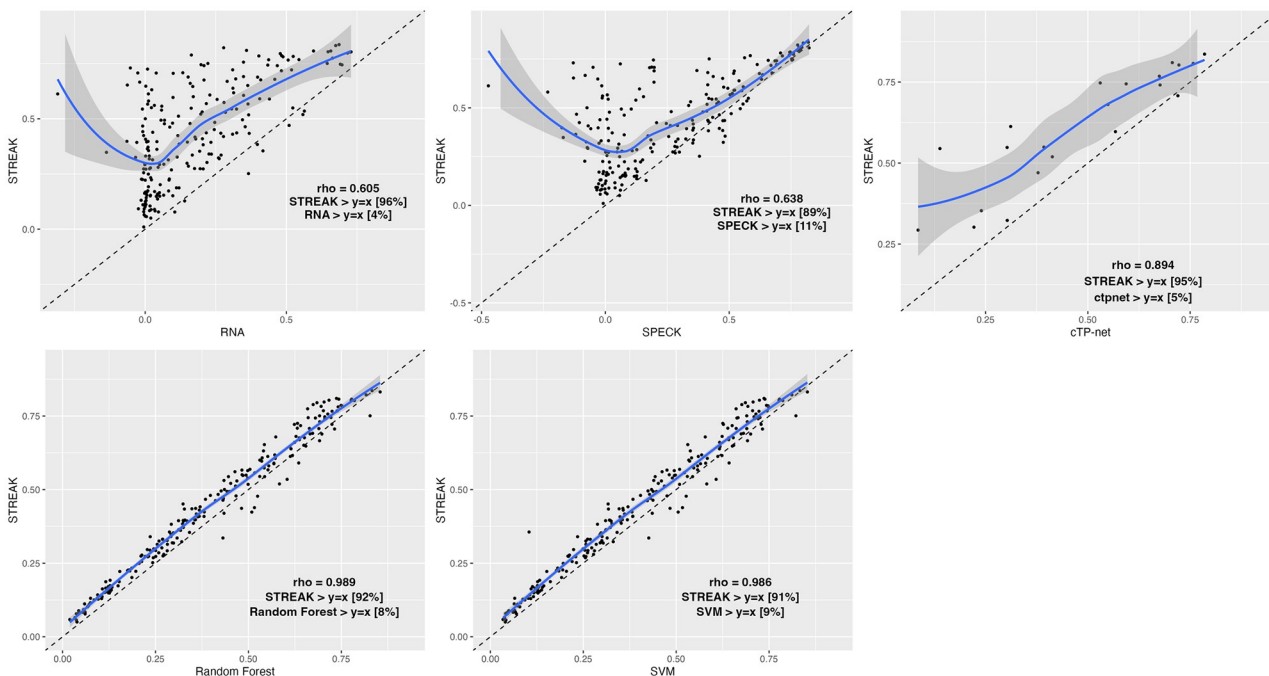

**Fig 8.** Correlation versus correlation scatter plots for the PBMC Hao data. Each point corresponds to a receptor from a maximum sample size of 217 receptors. LOESS (locally estimated scatterplot smoothing) function is applied to smooth out conditional means. Individual correlations are computed using the Spearman rank correlation metric.

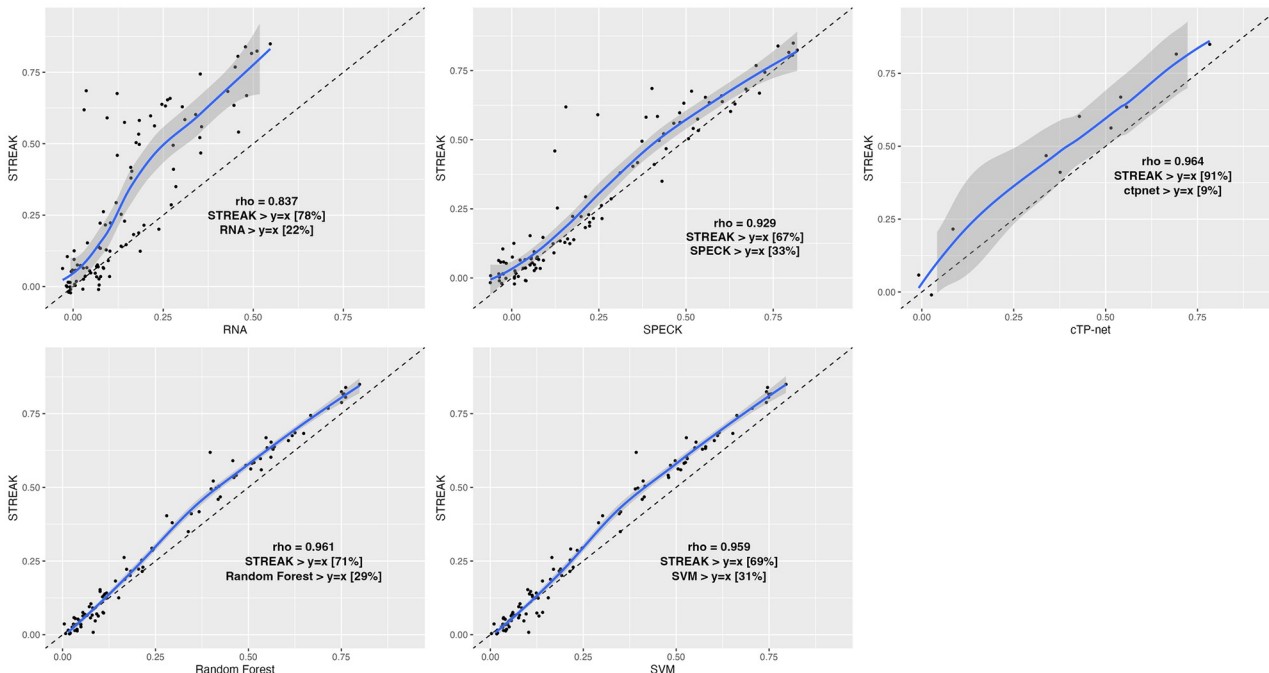

**Fig 9.** Correlation versus correlation (computed using the Spearman rank correlation metric) scatter plots for the Mouse Spleen data. Each point corresponds to a receptor from a maximum sample size of 107 receptors.

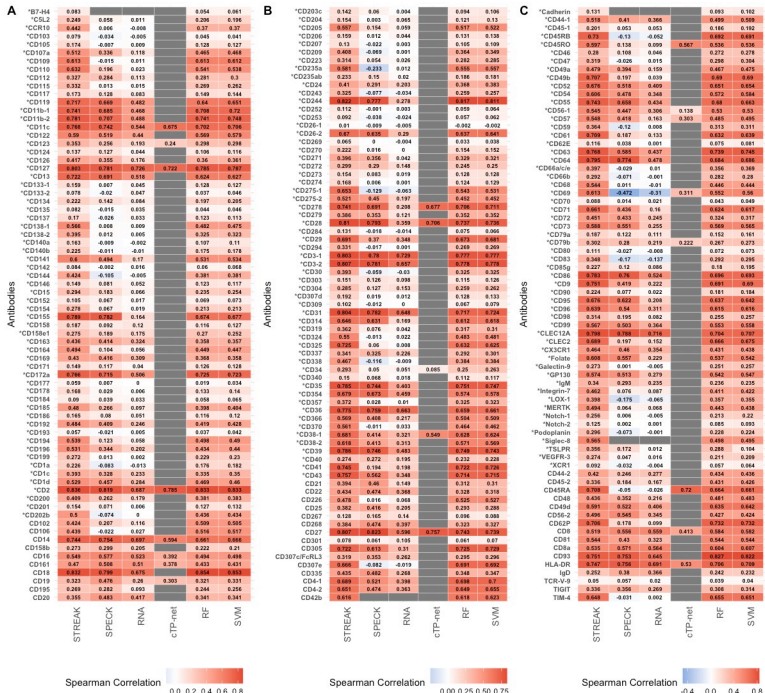

**Fig 10.** Average rank correlations between CITE-seq data and receptor abundance values estimated using STREAK and all comparative methods as evaluated using the 5-fold cross-validation approach for training data consisting of 12,000 cells from the Hao data. Asterisk text format is used to indicate receptors for which the STREAK estimate has highest correlation with corresponding CITE-seq data among all evaluated methods. Limits of the gradient color scale are determined by the minimum and maximum average correlation values for all comparative methods combined.

approach. To accomplish this, we evaluated STREAK against SPECK, the normalized RNA transcript, the RF model, and cTP-net using co-expression associations constructed from the joint scRNA-seq/CITE-seq Hao training data consisting of 5,000, 7,000, 10,000, 20,000 and 30,000 cells. Each set of trained associations were evaluated on five subsets of 5,000, 7,000 and 10,000 cells and a subset of 50,000 cells from the Unterman data. S7A–S7E Fig plot the proportion of receptors that have the highest rank correlations with CITE-seq data when abundance is estimated using STREAK, SPECK or the normalized RNA transcript with training performance on 5,000, 7,000, 10,000, 20,000 or 30,000 cells from the Hao data. Similarly, S8A–S8E Fig indicate the analogous proportion of receptors that have the highest rank correlations with CITE-seq data when estimated using STREAK and cTP-net while S9A–S9E Fig visualize the equivalent results for the RF model trained on co-expression associations learned from subsets of 5,000, 7,000, 10,000, 20,000 and 30,000 cells from the Hao data. These results demonstrate that the relative performance of STREAK is consistent across all training cell subsets, thereby underscoring STREAK's robustness to training data size for the cross-training evaluation strategy.

Our subsequent examination assessed STREAK's sensitivity to gene set size. We performed this analysis for the Hao data and the 5-fold cross-validation evaluation strategy using trained associations from the top 5, 10, 15, 20, 25 and 30 most correlated scRNA-seq transcripts with CITE-seq data. S10 Fig quantifies the proportion of receptors that have high rank correlations with CITE-seq data when estimated using STREAK versus when estimated using SPECK or the normalized RNA approach. Similarly, S11 Fig compares the proportion of receptors that

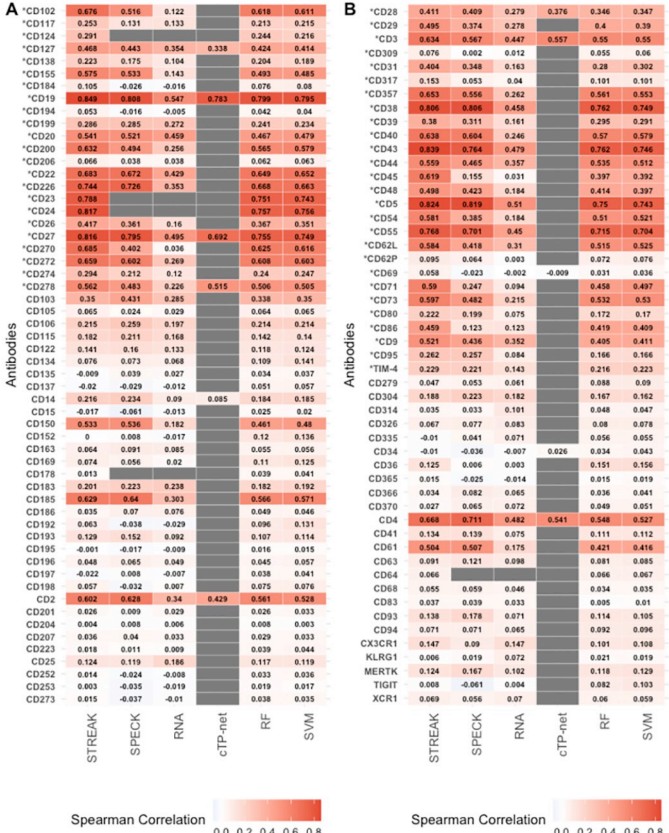

**Fig 11.** Average rank correlations between CITE-seq data and receptor abundance values estimated using STREAK and comparative methods as evaluated using the 5-fold cross-validation approach for training data consisting of 3,940 cells from the Mouse Spleen dataset.

have the highest correlation with the corresponding CITE-seq data when estimated using STREAK versus cTP-net while S12 Fig compares this proportion for STREAK versus the RF model. All three plots emphasize that STREAK is generally insensitive to gene set size and consistently performs better than comparative evaluation strategies given a gene set size range of 5 to 30 genes.

## Discussion

In this paper, we detail a novel supervised receptor abundance estimation method, STREAK, which functions by first learning associations between gene expression and protein abundance data using joint scRNA-seq/CITE-seq training data and then leverages these associations to perform thresholded gene set scoring on the target scRNA-seq data. We evaluate this method on six joint scRNA-seq/CITE-seq datasets representing four different tissue types and two organisms using two evaluation strategies, which include the human PBMC Hao, PBMC Unterman, MALT, Monocytes, MPEM and mouse Spleen and Lymph Nodes. We compare STREAK's performance against both unsupervised abundance estimation techniques such as SPECK and the normalized RNA approach and supervised methods such as cTP-net, RF, and SVM. This evaluation demonstrates that for the majority of the analyzed receptors, STREAK abundance estimates are more accurate than those produced alternative techniques, as assessed

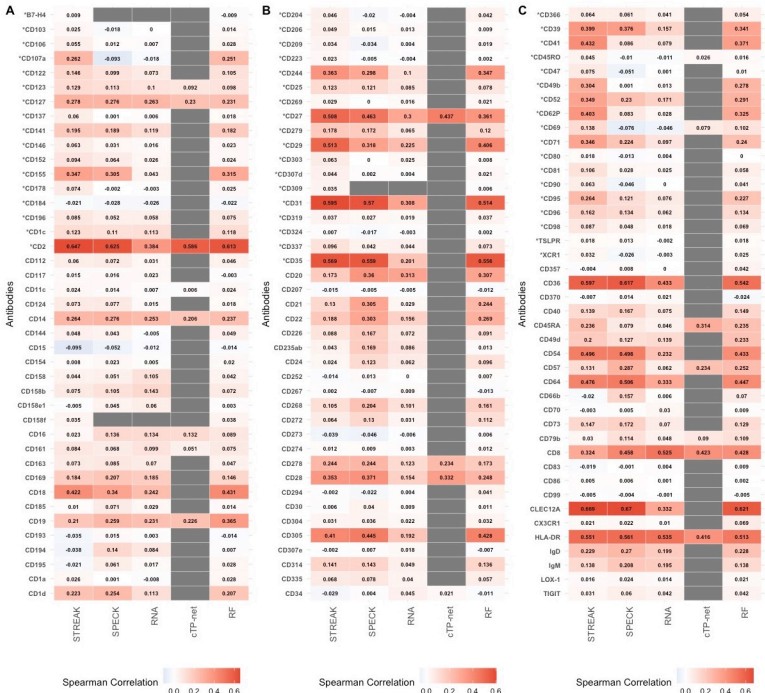

**Fig 12.** Average rank correlations between CITE-seq data and abundance values estimated using STREAK and comparative methods evaluated using the cross-training approach trained on a subset of 5,000 cells from the Hao data and evaluated on a subset of 50,000 cells from the Unterman data.

by the Spearman rank correlation between estimated abundance profiles and associated CITE-seq data.

A key strength of STREAK in comparison to neural network or ensemble-based supervised receptor abundance estimation techniques is that the weighted gene sets used for cell-level estimation are simple to interpret and customize. Researchers can easily add or remove genes and

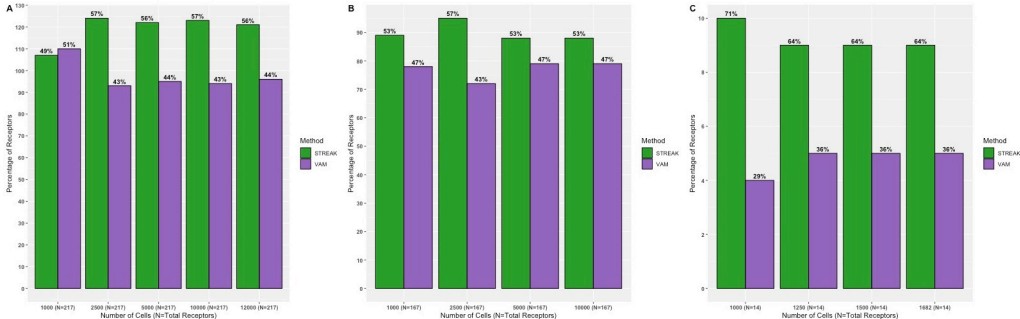

**Fig 13.** Gene set scoring versus thresholding sensitivity analysis examining frequency of receptors with highest average rank correlations between CITE-seq data and abundance values estimated using STREAK (i.e., estimation via gene set scoring followed by thresholding) or VAM (i.e., estimation using just gene set scoring) evaluated using the 5-fold cross-validation approach with the indicated training data ranging from 1,000 to 12,000 cells for the Hao data (Fig 13A) 1,000 to 10,000 cells for the Unterman data (Fig 13B) and 1,000 to 1,682 cells for the MALT data (Fig 13C).

adjust weights for empirically derived sets or define entirely new gene sets to reflect specific biological knowledge or better adapt to the expected pattern of gene expression in a given tissue type or environment.

One limitation of STREAK is that receptor gene set construction will typically require access to joint scRNA-seq/CITE-seq training data that is ideally measured on the same tissue type (and under similar biological conditions) as the target scRNA-seq data. A related limitation is that this gene set construction step is computationally expensive since it entails comparison of each CITE-seq ADT transcript with every gene in the scRNA-seq expression matrix. While both limitations can be avoided if users feel comfortable manually defining the receptor gene sets, we anticipate that access to experimental training data will be necessary to create accurate estimates for most receptor proteins. Our current R package implementation for STREAK is available on CRAN [39]. This package supports both the gene set construction and the receptor abundance estimation components of the STREAK algorithm. The gene set construction functionality can moreover be modified to compute co-expression associations using metrics other than the Spearman rank correlation.

In short, we outline a new supervised receptor abundance estimation method that leverages joint associations between transcriptomics and proteomics data to generate abundance estimates using thresholded cell-level gene set scores. The STREAK method produces more accurate abundance estimates relative to other unsupervised and supervised abundance estimation techniques with the potential to significantly improve the performance of downstream single cell analysis tasks such as cell typing/phenotyping and cell-cell signaling estimation.

## Supporting information

**S1 Fig.** Correlation versus correlation scatter plots for the PBMC Unterman data. Each point corresponds to a receptor from a sample size of 167 receptors.
(TIFF)

**S2 Fig.** Correlation versus correlation scatter plots for the MALT data. Each point corresponds to a receptor from a sample size of 14 receptors.
(TIFF)

**S3 Fig.** Correlation versus correlation scatter plots for the Monocytes data. Each point corresponds to a receptor from a sample size of 252 receptors.
(TIFF)

**S4 Fig.** Correlation versus correlation scatter plots for the MPEM data. Each point corresponds to a receptor from a sample size of 46 receptors.
(TIFF)

**S5 Fig.** Frequency of receptors with highest average Spearman rank correlations between CITE-seq data and abundance profiles estimated using STREAK, SPECK and normalized RNA approach or cTP-net or RF for the 5-fold cross-validation approach with training data ranging from 1,000 to 12,000 cells for the Hao data (S5A, S5D, S5G), 1,000 to 10,000 cells for the Unterman data (S5B, S5E, S5H) and 1,000 to 1,682 cells for the MALT data (S5C, S5F, S5I) and 5,000 cells from the Hao data for the cross-training evaluation approach (S5J, S5K, S5L). The horizontal axis for the 5-fold cross-validation evaluation plots (S5A-S5I) indicates the number of cells used for training while the horizontal axis for the cross-training evaluation plots (S5J-S5L) indicates the number of target cells evaluated from the Unterman data.
(TIFF)

**S6 Fig.** Percentage of receptors with highest average Spearman rank correlations between CITE-seq data and abundance profiles estimated using STREAK, SPECK and normalized RNA approach or cTP-net or RF for the 5-fold cross-validation approach with 5,000 cells from the Hao data for the cross-training evaluation approach (S6A, S6B, S6C). The horizontal axis for these plots (S6A-S6C) indicates the number of target cells evaluated from the Unterman data.
(TIFF)

**S7 Fig.** Training data sensitivity analysis examining frequency of receptors with highest average rank correlations between CITE-seq data and abundance values estimated using STREAK, SPECK and normalized RNA transcript evaluated using the cross-training approach with Hao training data consisting of 5,000 (S7A), 7,000 (S7B), 10,000 (S7C), 20,000 (S7D) and 30,000 (S7E) cells. The horizontal axis for each plot indicates the number of target cells evaluated from the Unterman data.
(TIFF)

**S8 Fig.** Training data sensitivity analysis examining frequency of receptors with highest average rank correlations between CITE-seq data and abundance values estimated using STREAK and cTP-net via the cross-training strategy.
(TIFF)

**S9 Fig.** Training data sensitivity analysis examining frequency of receptors with highest average rank correlations between CITE-seq data and abundance values estimated using STREAK and the RF model via the cross-training strategy.
(TIFF)

**S10 Fig.** Gene set size sensitivity analysis examining frequency of receptors with highest average rank correlations between CITE-seq data and abundance values estimated using STREAK, SPECK and normalized RNA transcript evaluated using the 5-fold cross-validation approach with the indicated training data ranging from 1,000 to 12,000 cells for the Hao data and gene set size consisting of 5, 10, 15, 20, 25 and 30 genes.
(TIFF)

**S11 Fig.** Gene set size sensitivity analysis between CITE-seq data and abundance profiles estimated with STREAK and cTP-net using the 5-fold cross-validation approach with training data ranging from 1,000 to 12,000 cells for the Hao data and gene set size consisting of 5, 10, 15, 20, 25 and 30 genes.
(TIFF)

**S12 Fig.** Gene set size sensitivity analysis between CITE-seq data and abundance profiles estimated with STREAK and the RF model using the 5-fold cross-validation approach with training data ranging from 1,000 to 12,000 cells for the Hao data and gene set size consisting of 5, 10, 15, 20, 25 and 30 genes.
(TIFF)

**S13 Fig.** Average rank correlations between CITE-seq data and receptor abundance values estimated using STREAK and comparative methods as evaluated using the 5-fold cross-validation approach for training data consisting of 12,000 cells from the Unterman dataset.
(TIFF)

**S14 Fig.** Average rank correlations between CITE-seq data and receptor abundance values estimated using STREAK and comparative methods as evaluated using the 5-fold cross-

validation approach for training data consisting of 1,682 cells from the MALT dataset.
(TIFF)

**S15 Fig.** Average rank correlations between CITE-seq data and receptor abundance values estimated using STREAK and comparative methods as evaluated using the 5-fold cross-validation approach for training data consisting of 7,422 cells from the Monocytes dataset.
(TIFF)

**S16 Fig.** Average rank correlations between CITE-seq data and receptor abundance values estimated using STREAK and comparative methods as evaluated using the 5-fold cross-validation approach for training data consisting of 994 cells from the MPEM dataset.
(TIFF)

## Acknowledgments

We would like to acknowledge the supportive environment at the Geisel School of Medicine at Dartmouth where this research was performed.

## Author Contributions

**Conceptualization:** Azka Javaid, Hildreth Robert Frost.

**Data curation:** Azka Javaid.

**Formal analysis:** Azka Javaid.

**Funding acquisition:** Hildreth Robert Frost.

**Investigation:** Azka Javaid, Hildreth Robert Frost.

**Methodology:** Azka Javaid, Hildreth Robert Frost.

**Software:** Azka Javaid.

**Supervision:** Hildreth Robert Frost.

**Validation:** Azka Javaid.

**Visualization:** Azka Javaid.

**Writing – original draft:** Azka Javaid, Hildreth Robert Frost.

**Writing – review & editing:** Azka Javaid, Hildreth Robert Frost.

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
