## [Decision Letter · Decision Letter 0]

21 Feb 2023

Dear Ms. Javaid,

Thank you very much for submitting your manuscript "STREAK: A Supervised Cell Surface Receptor Abundance Estimation Strategy for Single Cell RNA-Sequencing Data using Feature Selection and Thresholded Gene Set Scoring" for consideration at PLOS Computational Biology. As with all papers reviewed by the journal, your manuscript was reviewed by members of the editorial board and by several independent reviewers. The reviewers appreciated the attention to an important topic. Based on the reviews, we are likely to accept this manuscript for publication, providing that you modify the manuscript according to the review recommendations. In particular, all the reviewers gave useful comments to improve the overall presentation and results which could strengthen the work.

Sincerely,

Elena Papaleo, PhD

Academic Editor

PLOS Computational Biology

Mark Alber

Section Editor

PLOS Computational Biology

Reviewer's Responses to Questions

**Comments to the Authors:**

Reviewer #1: In this work, the authors present a supervised method to estimate receptor abundance on individual cells from scRNA-seq data, which is trained on joint scRNA/seq/CITE-seq data. The authors benchmark their method against other state-of-the-art methods designed to address the same problem and show that theirs works better.

The problem the authors are trying to solve is relevant given the recent interest in single-cell multiomics technologies, and the method is well explained and makes sense. However, I found the presentation of the actual results confusing:

• For example, several plots rely on the “proportion of abundance profiles that have the highest Spearman rank correlation” metric, which is a hard description to parse. Turns out that this is the number of receptors with the highest correlation with CITE-seq data *compared to the other estimation methods*, which explains trends that are otherwise puzzling like some of the bars going down with increasing numbers of cells in Fig 2. Also the main text uses the word “proportion” but the plots use “frequency”, and actually show absolute numbers in their y axes. If this metric is to be used, it should be explained better and with more consistent terminology.

• Furthermore, figs. 3-6 show that in several cases, even though STREAK has higher performance, the difference is not that large. This makes it questionable to use a metric based on which method has the highest correlation. In other words, there is a loss of information of the actual correlation values in figs. 2, 7, 8, and 9 that I think is important.

• The inclusion of the different numbers of cells in Figs 2, 7, and 9 is interesting in principle, but no insight is extracted or discussed in the text. Fig. 8 is the exception, where the different numbers of cells used for training have a clear purpose. But even then, the point of using different number of cells for their test dataset is unclear.

• Figs 3-6 contain tables with every estimation for every receptor in every dataset, which makes it hard to parse or extract any insights from them.

• Finally, the plot text (axis and tick labels) is too small, which makes it impossible to read on printed paper and requires lots of zooming in a computer.

I would recommend the authors to use correlation vs correlation scatter plots, with each point corresponding to a receptor and the x and y axis showing the spearman correlations between CITE-seq and each one of the estimation methods under study. The position of the dots can be compared with the x=y diagonal to see which method is overall better for most receptors, and numerical information about the correlation values is not lost. This makes for a more succinct comparison of correlation performance metrics, and maybe the number of figures and panels can be reduced. For examples, check Fig. 1b of https://www.nature.com/articles/s41592-021-01252-x.

Finally, while the selection of competing methods against which this work is compared is adequate, the authors avoid comparing against PIKE-R2P because, they claim, the code was not available. As of this writing the code seems to be available at https://github.com/JieZheng-ShanghaiTech/PIKE-R2P. Therefore, I would ask the authors to perform this comparison.

Reviewer #2: I read this paper entitled “STREAK: A Supervised Cell Surface Receptor Abundance Estimation Strategy for Single Cell RNA-Sequencing Data using Feature Selection and Thresholded Gene Set Scoring” with great interest, as it proposes a new solution to the challenges associated with estimating receptor abundance for scRNA-seq target data.

The authors have developed a new supervised receptor abundance estimation method called STREAK for single-cell transcriptomics data. This method is an improvement over their previous unsupervised method SPECK, as it leverages associations learned from joint scRNA-seq/CITE-seq training data and uses a thresholded gene set scoring mechanism. The authors evaluate STREAK against both unsupervised and supervised methods on three joint scRNA-seq/CITE-seq datasets and conclude that it outperforms other techniques and provides a more biologically interpretable and transparent statistical model.

I commend the authors for the introduction of a new and improved method for cell surface receptor abundance estimation. The use of joint scRNA-seq/CITE-seq training data to leverage associations and improve the accuracy of the estimation is one of the strengths. Besides, the authors have also carried out a detailed evaluation of the method on multiple datasets and a demonstration of its superiority over other techniques.

I have following questions/suggestions for the authors:

1. How logical and fair is the comparison with other techniques given those methods may have different assumptions and limitations, particularly the comparison between supervised and unsupervised techniques?

2. The evaluation of the method is limited to three joint scRNA-seq/CITE-seq datasets representing two different tissue types and using two evaluation strategies. In my opinion, a wider range of data and a larger number of datasets should be used to validate the generalizability and robustness of the method. Can we use or expand the same method to other tissue types and bigger datasets?

3. I would appreciate it if the authors can prepare and highlight the comparison of different methods in a Tabular format as a Main Table.

Reviewer #3: in the manuscript "STREAK: A Supervised Cell Surface Receptor Abundance Estimation Strategy for Single Cell RNA-Sequencing Data using Feature Selection and Thresholded Gene Set Scoring: the authors report on a new supervised receptor abundance estimation method.

The manuscript and the tool are of definite interest and I have only minor comments:

1. Would be possible to apply this tool to generate accurate estimation of cell surface receptor abundance for tissue-specific single cell data beyond blood / immune cells?

2. The authors evaluated STREAK against the Random Forest (RF) algorithm, but RF heavily depends on feature selection. Can the authors evaluate the data with at least one more machine learning algorithm such as SVM or GBM to confirm STREAK superiority to other machine learning methods.

3. The authors say that they did not compare STREAK against PIKE-R2P as it is not in CRAN, but the R code is available in github. Comparison should be conducted.

4. The CDF in the manuscript had gamma distribution, but single cell data usually follows a negative binomial distribution. Is there an advantage to use gamma distribution?

**Have the authors made all data and (if applicable) computational code underlying the findings in their manuscript fully available?**

Reviewer #1: **No: **Package can be found by searching on CRAN. I would recommend a direct link and a copy on github or similar if possible.

Reviewer #2: Yes

Reviewer #3: Yes

PLOS authors have the option to publish the peer review history of their article (what does this mean?). If published, this will include your full peer review and any attached files.

Reviewer #1: No

Reviewer #2: No

Reviewer #3: No

Figure Files:

Data Requirements:

Reproducibility:

References:

---

## [Decision Letter · Decision Letter 1]

20 Jun 2023

Dear Ms. Javaid,

Thank you very much for submitting your manuscript "STREAK: A Supervised Cell Surface Receptor Abundance Estimation Strategy for Single Cell RNA-Sequencing Data using Feature Selection and Thresholded Gene Set Scoring" for consideration at PLOS Computational Biology. As with all papers reviewed by the journal, your manuscript was reviewed by members of the editorial board and by several independent reviewers. The reviewers appreciated the attention to an important topic. Based on the reviews, we are likely to accept this manuscript for publication, providing that you modify the manuscript according to the review recommendations from Reviewer 1.

Sincerely,

Elena Papaleo, PhD

Academic Editor

PLOS Computational Biology

Mark Alber

Section Editor

PLOS Computational Biology

Reviewer's Responses to Questions

**Comments to the Authors:**

Reviewer #1: Thanks to the authors for addressing my comments. I still have one important concern and a few presentation issues:

• The authors mention that they were unable to run PIKE-R2P from its github repo. However, the main text still says “We do not perform comparison against PIKE-R2P since its implementation does not currently exist on the Comprehensive R Archive Network (CRAN) or on alternative code repository platforms”, which is untrue. The authors should update this statement and describe, briefly but precisely, what issues they had with the software.

• From the description of their algorithm in the methods section, is seems that the VAM method only uses X*_R but not A. This is important since the authors claim that one advantage of their method is that the gene weights can be interpreted, or manually tuned to reflect biological knowledge. Can the authors clarify where A is used?

• Minor clarification: In step 2 of their summary of VAM, M[,k] is supposed to be a column matrix with dimensions (m2, 1) but the matrix product is Xk^T*(Ig*sigma^2)^-1*X_k, which is (m2, g)^T*(g, g)* (m2, g) = (g, m2)*(g, g)*(m2, g), which doesn’t make sense for a matrix product. Is this wrong, or is there a typo?

• While I appreciate that the bar plots have been simplified, I’m concerned that some of them do not show all the bars that they should. For example the MALT dataset in Figure 2 does not include bars for the RNA and RF methods. From looking at the supplementary figures, it seems that their value is zero, but this is not obvious from the figures. Maybe the authors can modify the plots to make this more explicit, or at the very least include a note in the figure legend saying that all bars unshown correspond to a value of zero.

• The authors say that heatmaps use bold text to indicate receptors where STREAK estimates are higher than other methods. But this is 1) really hard to see, 2) from what I can tell there are several receptors in which STREAK performs best that are not bolded. Example: CD11b-2 in Figure 9a. Can you clarify?

• Minor: In Figure 13, the y axis is frequency instead of proportion.

Reviewer #2: I have carefully reviewed the revised manuscript titled "STREAK: A Supervised Cell Surface Receptor Abundance Estimation Strategy for Single Cell RNA-Sequencing Data using Feature Selection and Thresholded Gene Set Scoring" along with responses to my previous comments, and I recommend accepting it for publication in PLOS Computational Biology.

I appreciate the thoroughness with which the authors addressed my concerns and incorporated the suggested revisions. The revised manuscript now provides a clear and comprehensive description of the STREAK method for estimating cell surface receptor abundance in single-cell RNA-sequencing data.

Reviewer #3: No further comments

**Have the authors made all data and (if applicable) computational code underlying the findings in their manuscript fully available?**

Reviewer #1: Yes

Reviewer #2: Yes

Reviewer #3: Yes

PLOS authors have the option to publish the peer review history of their article (what does this mean?). If published, this will include your full peer review and any attached files.

Reviewer #1: No

Reviewer #2: **Yes: **Dr. Vikas Sharma

Reviewer #3: No

Figure Files:

Data Requirements:

Reproducibility:

References:

---

## [Editor Report · Decision Letter 2]

7 Aug 2023

Dear Ms. Javaid,

We are pleased to inform you that your manuscript 'STREAK: A Supervised Cell Surface Receptor Abundance Estimation Strategy for Single Cell RNA-Sequencing Data using Feature Selection and Thresholded Gene Set Scoring' has been provisionally accepted for publication in PLOS Computational Biology.

Best regards,

Elena Papaleo, PhD

Academic Editor

PLOS Computational Biology

Mark Alber

Section Editor

PLOS Computational Biology

---

## [Editor Report · Acceptance letter]

17 Aug 2023

PCOMPBIOL-D-22-01790R2 

STREAK: A Supervised Cell Surface Receptor Abundance Estimation Strategy for Single Cell RNA-Sequencing Data using Feature Selection and Thresholded Gene Set Scoring

Dear Dr Javaid,

I am pleased to inform you that your manuscript has been formally accepted for publication in PLOS Computational Biology. Your manuscript is now with our production department and you will be notified of the publication date in due course.

With kind regards,

Zsofi Zombor
